# Bosonic spinons in anisotropic triangular antiferromagnets

Youngsu Choi[1,2,4], Suheon Lee [1,4], Je-Ho Lee[1,4], Seungyeol Lee[1], Maeng-Je Seong [1✉] & Kwang-Yong Choi[1,3✉]

Anisotropic triangular antiferromagnets can host two primary spin excitations, namely, spinons and triplons. Here, we utilize polarization-resolved Raman spectroscopy to assess the statistics and dynamics of spinons in $Ca_3ReO_5Cl_2$. We observe a magnetic Raman continuum consisting of one- and two-pair spinon-antispinon excitations as well as triplon excitations. The twofold rotational symmetry of the spinon and triplon excitations are distinct from magnons. The strong thermal evolution of spinon scattering is compatible with the bosonic spinon scenario. The quasilinear spinon hardening with decreasing temperature is envisaged as the ordering of one-dimensional topological defects. This discovery will enable a fundamental understanding of novel phenomena induced by lowering spatial dimensionality in quantum spin systems.

[1] Department of Physics, Chung-Ang University, Seoul 06974, Republic of Korea. [2] Department of Energy Science, Sungkyunkwan University, Suwon 16419, Republic of Korea. [3] Department of Physics, Sungkyunkwan University, Suwon 16419, Republic of Korea. [4]These authors contributed equally: Youngsu Choi, Suheon Lee, Je-Ho Lee. ✉email: mseong@cau.ac.kr; choisky99@skku.edu

Quantum mechanics defines two fundamental classes of particles-bosons and fermions whose behaviors are dictated by exchange statistics[1,2]. Elementary, quasiparticles, and composite particles in high-energy and condensed matter physics are either bosons or fermions. However, it is an experimental challenge to probe the quantum statistics of spinons, which emerge from a quantum spin liquid. Spinons are fractionalized excitations with a fractional quantum number and lay a ground of conceptual developments in topological order and high-temperature superconductivity[3,4].

The $s = 1/2$ Heisenberg antiferromagnet on an anisotropic triangular lattice (ATL) constitutes an archetypical instance of dimensional reduction[5–13]. An ATL consists of intrachain interaction $J$ and interchain interaction $J'$. Competing interchain interactions $J'$ add frustration to a two-dimensional (2D) spin network. At zero temperature and in an asymptotically low-energy limit, individual spin chains are weakly coupled by fluctuation-generated interactions[14]. Physical ramifications of this dimensional reduction are the emergence of deconfined fractional excitations and gauge symmetry[15–18]. However, much remains enigmatic about a dynamical decoupling of the 2D magnet into a one-dimensional (1D) spin chain.

The magnetic phase of the ATL model is tuned by the anisotropy parameter $\xi = J'/J$. Singularly, a Tomonaga–Luttinger liquid (TLL) state appears over a wide parameter range of $\xi < 0.6$–$0.7$[8–14]. This regime is understood in the framework of a one-dimensionalization process. It is well known that the low-temperature physics of spin chains is described by a TLL[18]. Fractionalized spinons carrying spin-1/2 are created in pairs via a spin-flip and freely propagate in individual spin chains[19–21]. In the presence of the frustrating $J'$ couplings, spinon hopping is prohibited between neighboring chains as it costs large kinetic energy. Instead, it causes spinon pairs to form bound states (=triplons)[5,8]. The triplons-collective excitations with integer quantum number $S = 1$ can propagate coherently between chain subsystems. As such, ATLs in the intermediate $\xi$ range host two species of elementary excitations, i.e., deconfined spinons and bound spinons. Besides, theoretical calculations have unveiled the contrasting dynamical response between bosonic and fermionic spinons, thereby raising an exciting prospect of detecting the spinon statistics in ATLs[6].

In the quest for ATLs, $Cs_2CuCl_4$ and $Ca_3ReO_5Cl_2$ have been reported as benchmark materials[22–30]. Both compounds realize a similar degree of spatial anisotropy with $\xi = 0.32$–$0.34$ and, thus, are expected to fall in a TLL phase. However, small perturbations including Dzyaloshinskii–Moriya and interlayer interactions set in spiral orders at $T_N = 0.62$ K for $Cs_2CuCl_4$ and $T_N = 1.13$ K for $Ca_3ReO_5Cl_2$. Notably, the exchange energy scale $J = 3.5$ meV of $Ca_3ReO_5Cl_2$ is about ten times larger than that of $Cs_2CuCl_4$[28,30]. In addition to the sizable $J$, the large ratio $J/T_N \approx 35$ in $Ca_3ReO_5Cl_2$ provides a sufficiently wide temperature window to access the inherent low-energy physics of ATLs.

Here, we utilize a polarization-resolved Raman spectroscopy to characterize spinons and triplons with respect to temperature, symmetry, and dynamics. We observe one-pair and two-pair spinon–antispinon excitations dressed by triplons. The bosonic-like evolution of spinon scattering with temperature is interpreted as fingerprints of the dimensional reduction occurring in ATLs.

## Results

**Spinon and triplon excitations.** In Fig. 1a–c, we schematically illustrate the $S = 1/2$ spinons (pink-colored domains) and $S = 1$ triplons (green ovals) and their dispersions. We further sketch two scattering geometries on the ATL plane in Fig. 1d. For ($b\theta$) scattering configuration, the incoming light polarization $\varepsilon_{in} = (0, 1, 0)$

is fixed along the chain direction (the $b$ axis), while the scattered light polarization $\varepsilon_{out} = (0, \cos\theta, \sin\theta)$ is rotated by an angle $\theta$ relative to the $b$ axis. In ($\theta\theta$) configuration, the incident and outgoing light polarizations are parallel to rotating angle $\theta$.

Figure 1e, f presents the magnetic Raman response obtained after subtracting phonons in two representative ($\theta\theta$) polarizations ($\theta = 0°$ and $60°$). For phonon modes, we refer to Supplementary Note 1 and 2 and Supplementary Figs. 1–3. The magnetic continuum extends to 150 cm$^{-1}$ (=5.4 $J$) and displays a double-peak structure, indicative of the occurrence of two types of spin excitations. For quantitative analysis, we decompose the magnetic continuum into two components denoted by T (green shading) and S (pink shading). The S and T excitations are centered at 86 cm$^{-1}$ (=3.1 $J$) and 45 cm$^{-1}$ (=1.6 $J$).

In the Fleury–Loudon–Elliott theory, spin-number-conserving Raman scatterings ($\Delta S = 0$) create both one-pair (1P) and two-pair (2P) spinon-antispinon excitations out of spin liquids (see "Methods" section)[31–33]. For 1D spin chains, the Raman intensity is proportional to the spinon density of states, that is, $I_{1p}(\omega) \propto \delta(\omega - \epsilon_{1D}(\mathbf{k}))$ for the 1P spinon excitations and $I_{2p}(\omega) \propto \delta(\omega - \epsilon_{1D}(\mathbf{k}) - \epsilon_{1D}(\mathbf{k}'))$ for the 2P. Here, the 1D spinon obeys the des Cloizeaux–Pearson dispersion $\epsilon_{1D}^{low}(\mathbf{k}) = (\pi J/2)|\sin ka|$[21], giving a lower boundary of the spinon continuum (see Fig. 1c). Notably, the peak energy $\omega_{2p} = 3.1 J$ of the S excitations amounts to $2\epsilon_{1D}^{low}(\mathbf{k} = \pi/2) = \pi J$. The high-energy cutoff of 5.4 $J$ is close to $2\pi J$, corresponding to the upper boundary of $2\epsilon_{1D}^{upper}(\mathbf{k}) = 2\pi J|\sin(\frac{k}{2}a)|$. Given that the S continuum is nicely described by four-spinon kinematics, two pairs of spinon–antispinon excitations dominate the higher-energy magnetic continuum. As the spin chains are frustrated in the low-temperature limit ($T < J$), the spinon excitations may inherit from the TLL physics. Here, we recall that the quasi-1D antiferromagnet $KCuF_3$ with $T_N = 39$ K features a multi-spinon continuum centered around $\omega_{2p} = 2.6 J$ in the intrachain polarization[34], comparable to 3.1 $J$ of $Ca_3ReO_5Cl_2$. In the interchain polarization, however, the spinon continuum vanishes and only a sharp transverse magnon is visible. However, little is known about what extent the ATL spinons resemble the 1D spinons. Before proceeding, we mention that magnons survive only at temperatures below $T_N = 1.13$ K and, thus, the magnon contribution is negligible (vide infra).

We next turn our attention to the lower-energy T excitation, which has the maximum at 1.6 $J$ and extends up to ~5 $J$. Since its peak energy is half the 2P spinon–antispinon continuum and is close to $\epsilon_{1D}^{low}(\mathbf{k} = \pi/2) = \pi J/2$, the T component seems to pertain to the 1P spinon excitation. As compared in Fig. 1e, f, however, the T excitation gains a spectral weight against the S excitation as the polarization is rotated away from the chain direction. This gives a convincing rationale that the S and T excitations are of largely distinct origin. As discussed below, the T spin excitations mainly comprise triplons. Our interpretation is supported by the inelastic neutron scattering data that show spinonlike continuum and sharply dispersive modes arising from bound spinon pairs[30]. However, we note that in a triplet ($\Delta S = 1$) sector, the triplons are localized around the energy of $\epsilon_{1D}^{low}(\mathbf{k} = \pi/2) = \pi J/2$. In sharp contrast, the triplon excitations in a singlet ($\Delta S = 0$) sector exhibit a broad spectrum whose high-energy cutoff exceeds $\pi J$. This is partly due to the fact that the T excitations are composed of one and two triplons, resulting from the 1P and 2P spinon–antispinon confinements (Supplementary Note 3). Given the small binding energy $J'/2$ of the triplons (see below and Fig. 4d), on the other hand, the T contribution contains thermally deconfined spinons in the measured temperature range of $T = 4.3$–300 K, leading to a broadened spectrum.

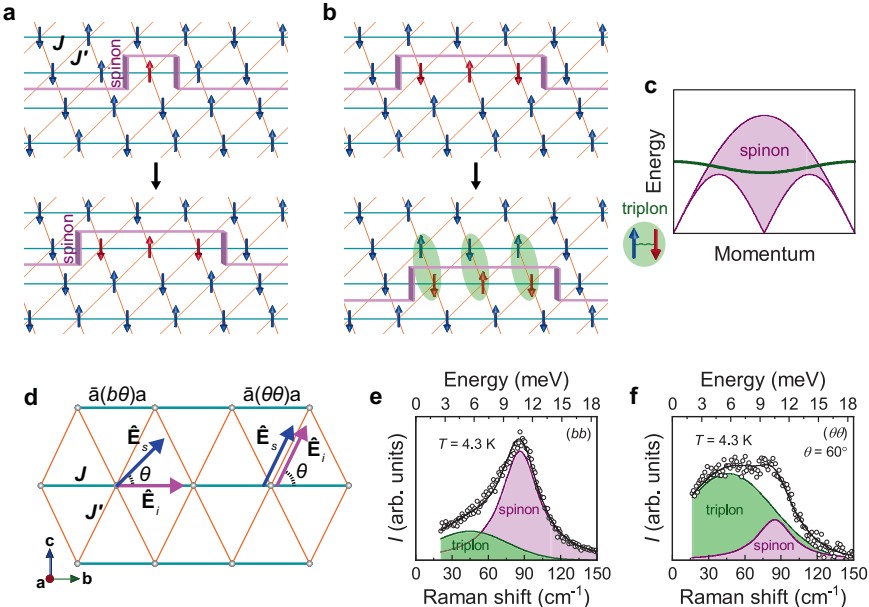

**Fig. 1 Schematic illustration of spinons and triplons in anisotropic triangular antiferromagnets. a** Flipping a local spin (red arrows) creates two domain walls (pink fold), corresponding to spinons carrying $S = 1/2$. The two spinons propagate freely along the chain direction. **b** The spinons can hop between the chains by forming their bound states (green oval). Pairs of the confined spinons constitute triplons carrying $S = 1$. **c** Dispersions of the spinon continuum (pink shading) and triplon (green line). **d** Sketch of $(b\theta)$ and $(\theta\theta)$ polarization configurations for angle-resolved Raman experiments. **e, f** Representative magnetic Raman continuum in $(\theta\theta)$ polarization with $\theta = 0°$ and $60°$. The magnetic continuum is decomposed into two spin excitations: triplons (green shading denoted by T) and two spinon-antispinon pair excitations (pink shading denoted by S). The triplon and spinon contributions are modeled by Gaussian and Lorentzian line profiles, respectively.

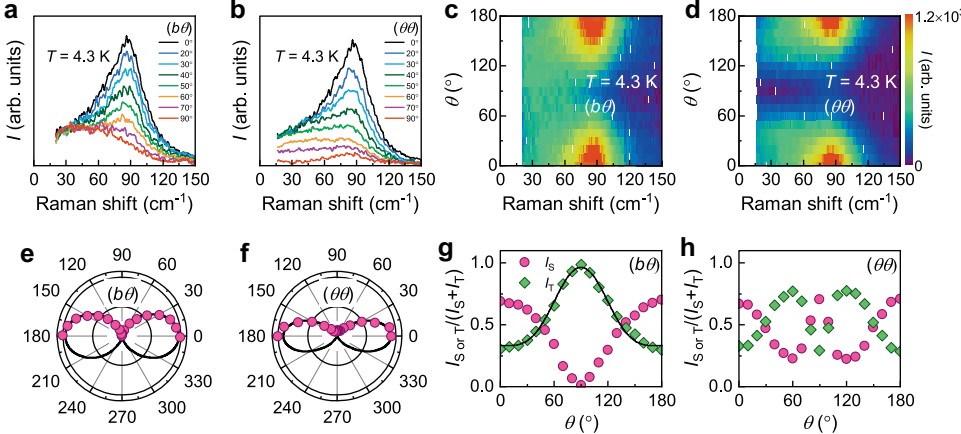

**Fig. 2 Angular dependence of magnetic excitations in two polarizations. a, b** Angular dependence of magnetic excitations measured at $T = 4.3$ K in $(b\theta)$ and $(\theta\theta)$ polarization configurations. Phonon modes are subtracted to obtain the magnetic continuum. **c, d** Color plots of the magnetic Raman scattering intensity in the angle-Raman-shift plane in $(b\theta)$ and $(\theta\theta)$ polarizations. **e, f** Polar plots of the integrated intensity of a two-pair spinon–antispinon continuum. The solids lines are fits to a $\cos^2\theta$ dependence, obeying the $A_1$ symmetry. **g, h** Relative intensity of the triplon $I_T/(I_S + I_T)$ and spinon $I_S/(I_S + I_T)$ components as a function of angle. The solid line is modeled to a function of $\sin^4\theta +$ constant.

**Rotational symmetry of spinon and triplon excitations**. Having identified the two different elementary excitations, we discuss their rotational symmetry. Figure 2a, b and Supplementary Figs. 4–5 exhibit the angle-dependent variation of the magnetic Raman spectra at $T = 4.3$ K and $T = 300$ K. In the high-$T$ hydrodynamic regime, we observe a quasielastic response originating from spin and energy fluctuations (Supplementary Note 6). In the low-$T$ TLL regime, the quasielastic scattering evolves to the well-defined spinon and triplon excitations as assigned above.

For both $(b\theta)$ and $(\theta\theta)$ polarizations, the magnetic scattering intensity $I_{tot}(\omega, \theta)$ is the maximum at $\theta = 0°$, i.e., in the intrachain

polarization. As the angle is increased toward $\theta = 90°$, $I_{tot}(\omega, \theta)$ continuously decreases, reaching its minimum at the perpendicular direction to the chain. Shown in Fig. 2c, d is the angle dependence of $I_{tot}(\omega, \theta)$ in the color plots. The $T = 4.3$ K magnetic continuum shows apparently twofold ($C_2$) rotational symmetry emulating the spatial anisotropy of the ATL lattice (the polar plots in Fig. 2e, f). Contrarily, lattice vibrations exhibit two-fold or four-fold symmetries (Supplementary Note 2 and Supplementary Figs. 2 and 3).

We note that the $C_2$ symmetry has the irreducible representations $A_1 + A_2$. We quantify the angle dependence of the spinon $I_S(\omega, \theta)$ and the triplon $I_T(\omega, \theta)$ contributions separately. Evident

from Supplementary Figs. 5 and 6, the spectral shape of both $I_T(\omega, \theta)$ and $I_S(\omega, \theta)$ is independent of the angle and polarization. The rotationally invariant spectral form validates the decomposition procedure. In addition, we observe a systematic suppression of $I_S(\omega, \theta)$ with increasing $\theta$ to 90°. This tendency can be deduced from the angular-dependent Raman operator $R^{(b\theta)}$ (see "Methods" section). At $\theta \neq 90°$, the magnetic Raman intensity $I^{(b\theta)}(\omega, \theta) \sim |R^{(b\theta)}|^2$ includes both $J$ and $J'$ terms, while $I^{(bc)}(\omega, \theta = 90°)$ contains only the $J'$ term. As such, the triplon scattering process is solely allowed in $(bc)$ polarization, if small perturbation interactions are neglected. Further, we note that the polar plots of the $(b\theta)$ data unveil that the spinons have the $A_1$ symmetry and follow a $\cos^2\theta$ dependence, whereas the triplons have the $A_1 + A_2$ symmetry (Supplementary Note 3). These angular dependencies are incompatible with the $\sin^2\theta$ dependence expected for two-magnon excitations[34,35]. This lends credence to our assignment that the observed magnetic continuum precludes a magnon contribution.

In Fig. 2g, h, we further plot the relative intensity as a function of angle, defined as $I_S(\theta)/I_{tot}(\theta)$ and $I_T(\theta)/I_{tot}(\theta)$. The $(b\theta)$ and $(\theta\theta)$ polarization channels show a contrasting behavior. In $(b\theta)$ polarization, $I_T(\theta)/I_{tot}(\theta)$ shows a $\sin^4\theta$-like increase with increasing $\theta$ up to 90°. In $(\theta\theta)$ polarization, however, $I_T(\theta)/I_{tot}(\theta)$ reaches the maximum at $\theta \approx 60°$, i.e., the direction of the interchain interaction. At the angle between 60° and 120°, triplon hopping is repressed because the two competing interchain interactions enhance quantum fluctuations. This reflects the geometrical form factor of an anisotropic spin Hamiltonian.

**Temperature evolution of magnetic excitations.** To trace dynamical behaviors of the triplons and spinons, we measured a thermal evolution of the magnetic continuum from $T = 4.3$ to 300 K in three $(bb)$, $(cc)$, and $(bc)$ polarizations. As shown in Fig. 3a–c, the parallel-polarization magnetic continuum systematically softens and eventually melts into a quasielastic response at the temperature scale of $J$. This feature is qualitatively different from the cross-polarization magnetic continuum whose intensity gradually decreases, only developing weak quasielastic scattering at high temperatures. The polarization-dependent behavior suggests that disparate quasiparticles dictate spin dynamics between parallel and cross polarizations.

We further visualize this trend in the color plots of the magnetic Raman intensity vs. temperature (Fig. 3d–f). In parallel polarizations, two regimes are delimited by the crossover temperature set by the intrachain interaction $J$. As plotted in Fig. 3g, the peak energy $\omega_{2p}$ of the 2P spinon–antispinon continuum undergoes a quasilinear red-shift with increasing temperature. In $(bb)$ polarization, the slope of $\omega_{2p}$ alters cross $T_{SO} = 27$ K, the onset of short-range ordering (Supplementary Note 6 and Supplementary Fig. 9). In $(cc)$ polarization, a slope change takes place between $J'$ and $T_{SO}$. The peak energy softening at the temperature $T = J$ amounts to 30% of $\omega_{2p}$ ($T = 4.3$ K). This is not compatible with the thermal characteristics of spinons in spin liquids, which show only thermal damping of the low-energy spectral weight[36–38]. Rather, this behavior is highly reminiscent of the renormalization and damping of magnons in conventional magnets[33]. As the 2P spinon–antispinon continuum appears for temperatures below $J$, the quasi-linear development of 1D topological defects is interpreted as the physical manifestation of a dimensional reduction. According to theoretical calculations[6], the temperature dependence of the spinon continuum in ATLs conveys information about quantum statistics, which affects the many-body spinon density of states. In the case of fermionic spinons, the dynamical response undergoes little change with temperature. It is well established

that a Kitaev honeycomb model harbors Majorana fermions. The peak energy of a Majorana spinon continuum remains unchanged with temperature[38]. Contrarily, the bosonic spinons in ATLs are subject to their spectral shift to lower energies with increasing temperature[6]. In this regard, we conclude that the 30% softening of the spinon peak energy supports the bosonic character of ATL spinons.

A close look at the $(bb)$ and $(cc)$ magnetic spectra discloses that the quasielastic response is more prominent in the interchain polarization. As the interchain spectra involve mainly spin excitations between the chains, the increased spatial dimensionality explains enhanced diffusive scatterings. To examine this difference, we calculate the integrated intensity $I(T)$ of the magnetic continuum in two different energy intervals; (i) the 1P spinon–antispinon energy range of $\omega = 30$–$60$ cm$^{-1}$ and (ii) the 2P spinon–antispinon energy range of $\omega = 70$–$100$ cm$^{-1}$. The temperature dependence of the normalized $I_{1p}(T)$ and $I_{2p}(T)$ is plotted in Fig. 3h, i. On heating through $T_{SO}$, the $(cc)$ polarization $I_{1p}(T)$ (cyan pentagon) grows steadily in a whole $T$ range. The $(cc)$ polarization $I_{2p}(T)$ first drops with increasing temperature to 50 K and then becomes $T$-independent. As to $(bb)$ polarization, $I_{1p}(T)$ resembles a static magnetic susceptibility (Supplementary Fig. 9), whereas $I_{2p}(T)$ decays exponentially with the characteristic temperature of 30 K, somewhat smaller than $J$.

**Temperature evolution of one and two triplons.** Unlike the $(bb)$ and $(cc)$ polarization data, both $I_{1p}(T)$ and $I_{2p}(T)$ in $(bc)$ polarization decrease with temperature in the same fashion, modeled by a double-exponential form $\exp(-T/\triangle_1) + \exp(-T/\triangle_2)$ with $\triangle_1 = 13.3 - 15.1$ K and $\triangle_2 = 92.7 - 100.0$ K. The gap energy of $\triangle_1$ matches well with $J' = 13.6$ K, while the second gap energy of $\triangle_2$ is linked to a crossover to a 2D hydrodynamic regime (Supplementary Note 6 and Supplementary Fig. 9). The exponential growth of the scattering intensity is an unambiguous signature of the resonance-like scattering of bound states[39], confirming the triplon scenario. In this light, $\triangle_1$ is associated with the triplon gap, yet $\triangle_2$ should be taken as the phenomenological parameter of describing the altering dynamical response at high temperatures. Noticeably, the evaluated $\triangle_1$ is much smaller than the triplon peak energy of 45 cm$^{-1}$ ($\approx 65$ K). The broad triplon spectra as shown in Figs. 1e and 4c, d suggest that the triplons are dressed by the spinons and are subject to many-body interactions in a $\Delta S = 0$ Raman scattering process. As such, the dressed triplons are responsible for an underestimation of $\triangle_1$.

To scrutinize triplon dynamics and kinetics, we subtract the weak 2D diffusive signal from the raw spectra, $I_T(\omega, T) - I_T(\omega, T = 300 \text{ K})$. The results are plotted in Fig. 4. Evident from Fig. 4a, b, the magnetic excitations grow rapidly toward 0 K. We can decompose the magnetic excitations into two Gaussian profiles (Fig. 4c), denoted by 1 T and 2 T. Figure 4d exhibits a temperature dependence of the peak energy, linewidth, and normalized intensity of the 1 T (green inverse triangles) and 2 T (pink circles) excitations. For temperatures above $T = J'$, their peak energy and linewidth hardly vary with temperature. On cooling, the normalized intensity follows a Curie-like increase. A substantial softening and narrowing by 5 cm$^{-1}$ is seen for temperatures below $T = J'$. Concomitantly, the normalized intensity tends to saturate as the temperature approaches 0 K. The energy of a triplon bound mode is given by $\epsilon_{TB} = 2\epsilon_T - \epsilon_{BE}$, where $\epsilon_T$ is the energy of a bare triplon and $\epsilon_{BE}$ is the triplon binding energy. With lowering temperature below $T = J'$, $\epsilon_{BE}$ increases against thermal energy, thereby leading to the peak-energy softening. The binding energy of the triplons $\epsilon_{BE} = 7$K ($\approx J'/2$) is estimated therefrom. This corroborates that the frustrated interchain interactions are responsible for the formation of a spinon-bound state. All in all,

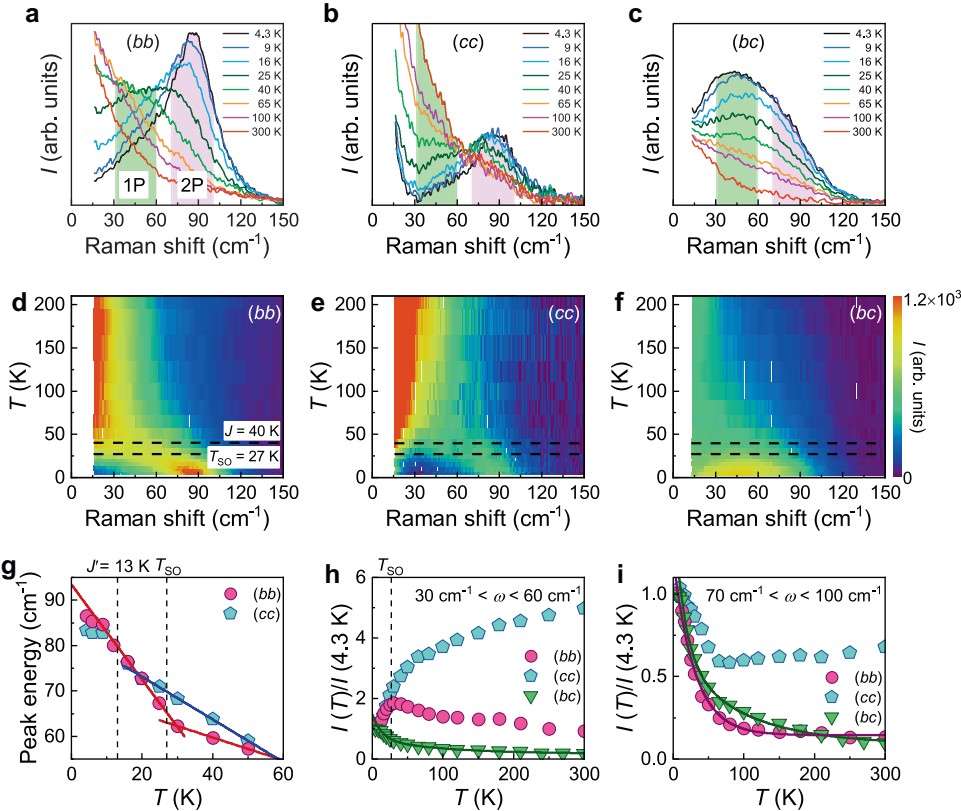

**Fig. 3 Temperature evolution of magnetic excitations. a–c** Temperature dependence of the magnetic Raman continuum in ($bb$), ($cc$), and ($bc$) polarizations. The green and pink shadings mark the characteristic energy scales of one-pair and two-pair spinon–antispinon excitations. **d–f** Color plots of the magnetic Raman intensity in the $T$-$\omega$ plane. The horizontal dashed lines indicate the onset of short-range ordering $T_{SO} = 27$ K and the intrachain interaction $J$. **g** Peak energy of the two-pair spinon–antispinon continuum as a function of temperature for ($bb$) and ($cc$) polarizations. The solid lines are linear guides to the eye. The vertical dashed lines mark $T_{SO}$ and the interchain interaction $J'$. **h, i** Temperature dependence of the integrated magnetic Raman intensity over the one-pair spinon-antispinon energy of $\omega = 30$-$60$ cm$^{-1}$ and the two-pair spinon-antispinon energy of $\omega = 70$-$100$ cm$^{-1}$. The solid lines are fits to an exponential function.

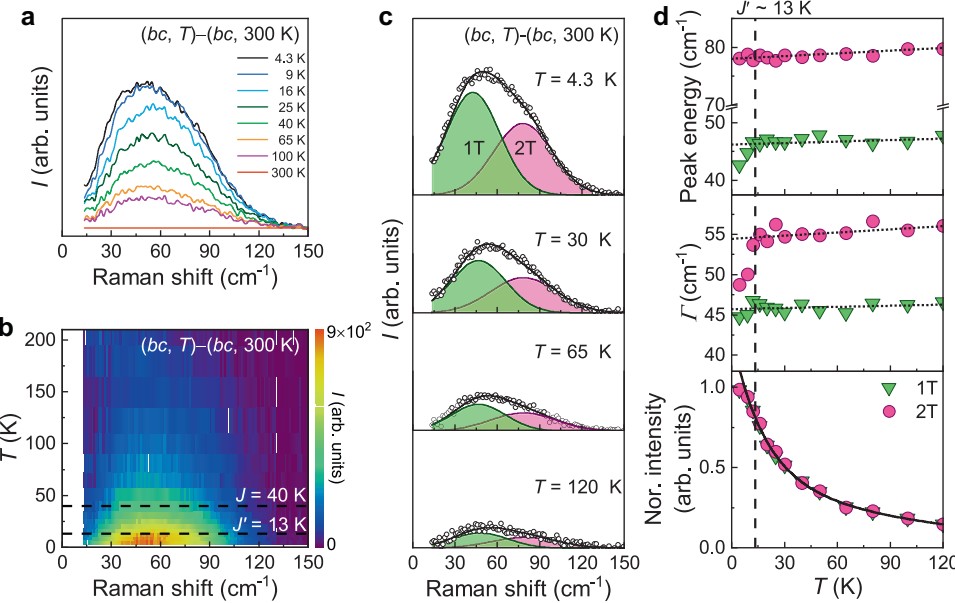

**Fig. 4 Temperature evolution of one and two triplons in ($bc$) polarization. a** Triplon signals in ($bc$) channel after subtraction of the $T = 300$ K spectrum. **b** Color plot of the subtracted triplon excitations in the $T$-$\omega$ plane. **c** Decomposition of the subtracted triplon excitations into one-triplon and two-triplon components at selected temperatures $T = 4.3, 30, 65$, and $120$ K. **d** Temperature dependence of the peak energy, linewidth, and normalized intensity of the one (green inverse triangles) and two (pink circles) triplons. The triplons soften and narrow for the temperatures below $T = J'$ as marked by the vertical dashed line. The solid line is fit to the intensity as described in the main text. The dotted linear are guides to the eye.

the $(bc)$ Raman data showcase a thermal crossover of paraspinons to the triplons in the temperature scale of $J'$. Therefore, the low-temperature 1 T and 2 T excitations are assigned to one-triplons and two-triplons, respectively. It should be cautioned that the 1 T and 2 T contain thermally deconfined spinons due to their small binding energy.

## Discussion

Based on polarization-dependent and temperature-dependent dynamical Raman scatterings of paradigmatic ATL $Ca_3ReO_5Cl_2$, we draw a landscape of low-energy excitations concerning their rotational symmetry, statistics, and dynamics.

In a low-$T$ TLL state, a magnetic Raman continuum is made of one-pair and two-pair spinon–antispinon excitations. It turns out that triplon occupies the majority of the one-pair spinon–antispinon spectral weight. In the intrachain polarization, Raman scatterings probe two species of spinons (at $3.1J$) and triplons (at $1.6J$), consistent with the Raman operator (see "Methods" section). This suggests that the ATL spinons are intrinsically dressed by triplons. The concurrence of the spinon and triplons lacks in 1D spin chains and reflects the dimensionality reduction.

Besides, the spinon and triplon excitations for both parallel and cross polarizations exhibit twofold rotational symmetry, being consistent with the $C_2$ symmetry of the underlying ATL lattice. The observed $\cos^2\theta$ dependence is incompatible with the $\sin^2\theta$ dependence predicted for two-magnon excitations. As the temperature is increased through $T = J$, the spinons in a low-$T$ TLL regime evolve to a quasielastic scattering in a high-$T$ hydrodynamic regime. The salient feature is that the spectral weight of the spinon continuum undergoes a quasilinear, substantial softening with increasing temperature, which is unexpected for 1D and 2D spinons. The thermal renormalization and damping of the spinons are in support of a bosonic spinon scenario. Heuristically, an analogy can be made with a Goldstone-like boson arising from spontaneous symmetry breaking. A dimensional reduction may be regarded as a dynamical process of symmetry lowering in spatial dimensionality.

In summary, a polarization-resolved Raman scattering study of $Ca_3ReO_5Cl_2$ allows characterizing the traits of emergent quasiparticles in ATLs. Our polarization and temperature dependence of magnetic excitations unveils the symmetry of dynamics of spinons and triplons, distinct from what is expected for magnons. The bosonic-like evolution of spinon scattering with temperature raises the question as to whether a bosonic spinon is a natural consequence of lowering spatial dimensionality in a TLL state, calling for developing a theoretical framework for an in-depth understanding of dimensional reduction in quantum spin systems.

## Methods

**Crystal growth and magnetic characterization.** Single crystals of $Ca_3ReO_5Cl_2$ were synthesized by a flux method. All chemicals were handled in an argon-filled glove box. CaO, $ReO_3$, and $CaCl_2$ were mixed in a molar ratio of 3:1:5 using an agate mortar. The mixture was pressed into a pellet and placed in a gold tube. The gold tube containing the starting materials was sealed in an evacuated quartz ampule. The ampule was heated at 1000 °C for 24 h and then slowly cooled down to 800 °C at a rate of 1 °C per hour. The unreacted starting materials and the excess $CaCl_2$ flux were washed out with distilled water. The obtained single crystals have a typical size of 2 mm × 2 mm × 0.5 mm with a shiny dark yellow surface.

Powder X-ray diffraction experiments on the ground single crystals confirmed that the crystal structure is identical to the previous report[39]. Laue X-ray diffraction enabled us to identify the flat surface of the as-grown crystals to the $bc$ plane. We measured dc magnetic susceptibility with a SQUID (Quantum Design MPMS) for B (=7 T)//b and B (=7 T)//c in the temperature range of $T = 2$–300 K (Supplementary Note 6 and Supplementary Fig. 9).

**Raman scattering measurements.** Raman scattering experiments were performed in exact backscattering geometry with a single-grating spectrometer (Princeton Instruments, SP-2500i) and a charge-couple device (CCD) detector. We used a 532 nm laser as an excitation source with a spot size of approximately ~1 μm on the sample in a liquid helium flow cryostat. The incident laser light passed a linear polarizer and a half-wave plate to rotate its polarization direction with respect to the crystal orientation of the sample. A Bragg grade notch filter (BNF) was used for spectral narrowing of the incident light. Rayleigh scattering was substantially eliminated by using additional BNFs, enabling the observation of Raman signal down to ~10 cm$^{-1}$. Another linear polarization analyzer followed by another half-wave plate was used for the emitted photons from the sample to ensure that the polarization direction of the emitted light entering the spectrometer remained the same, regardless of the direction of the analyzer, ruling out any possible polarization dependence of the spectrometer and the detector.

Temperature-dependent Raman scattering measurements were carried out in backscattering geometry with the excitation line $\lambda = 532$ nm of DPSS SLM laser. We collected the scattered spectra using a micro-Raman spectrometer (XperRam200VN, NanoBase) equipped with an air-cooled charge-coupled device (Andor iVac Camera). We employed a notch filter to reject Rayleigh scattering to a lower cutoff frequency of 15 cm$^{-1}$. The laser beam with $P = 80$ μW was focused on a few-micrometer-diameter spot on the surface of the crystals using a × 40 magnification microscope objectives. The samples were mounted onto a liquid-He-cooled continuous flow cryostat while varying a temperature between 4.3 and 300 K.

## Raman data analysis

*Analysis of one-pair and two-pair spinons.* According to the Loudon–Fleury (LF) approach, the Raman scattering operator is given by

$$R = \sum_{i,d_\mu} \left(\varepsilon_{in} \cdot d_\mu\right)\left(\varepsilon_{out} \cdot d_\mu\right) J_\mu S_i \cdot S_{i+d_\mu}, \tag{1}$$

where $d_\mu$ denotes the basis vectors of an anisotropic triangular lattice: $d_1 = (1,0), d_2 = \left(\frac{1}{2}, \frac{\sqrt{3}}{2}\right)$, and $d_3 = \left(-\frac{1}{2}, \frac{\sqrt{3}}{2}\right)$. $J_\mu$ defines the Heisenberg exchange interactions on the bond $d_\mu$. With polarizations of incoming and outgoing light, magnetic Raman scattering probes spin excitations via a two-spin-flip process. In a spin liquid, the LF scattering operator is expressed in terms of spinon operators $R \propto f_{i\alpha}^\dagger f_{i+d\sigma}^\dagger f_{i+d\sigma'} f_{i\sigma'}$. Accordingly, two pairs of a spinon-antispinon continuum are primarily excited. In addition, a one-pair spinon $R \propto f_{i\alpha}^\dagger f_{i+d\sigma} + f_{i+d\sigma}^\dagger f_{i\sigma}$ can be created.

For $(b\theta)$ polarization, the LF scattering operator is given by

$$R^{(b\theta)} = \sum_i \left\{ J\cos\theta\left(S_i \cdot S_{i+d_1} + S_i \cdot S_{i-d_1}\right) + \frac{J'}{4}(\cos\theta + \sqrt{3}\sin\theta)\left(S_i \cdot S_{i+d_2} + S_i \cdot S_{i-d_2}\right) \right.$$
$$\left. + \frac{J'}{4}(\cos\theta - \sqrt{3}\sin\theta)\left(S_i \cdot S_{i+d_3} + S_i \cdot S_{i-d_3}\right) \right\}. \tag{2}$$

For $(\theta\theta)$ polarization, the LF scattering operator is expressed as

$$R^{(\theta\theta)} = \sum_i \left\{ J\cos^2\theta\left(S_i \cdot S_{i+d_1} + S_i \cdot S_{i-d_1}\right) + \frac{J'}{4}(\cos\theta + \sqrt{3}\sin\theta)^2\left(S_i \cdot S_{i+d_2} + S_i \cdot S_{i+d_2}\right) \right.$$
$$\left. + \frac{J'}{4}(\cos\theta - \sqrt{3}\sin\theta)^2\left(S_i \cdot S_{i+d_3} + S_i \cdot S_{i-d_3}\right) \right\}. \tag{3}$$

*Analysis of the angular dependence of Raman intensity.* An anisotropic triangular lattice possesses $C_{2v}$ symmetry having the irreducible representations $A_1 + A_2$. Considering the material absorbs incident light, the Raman tensors of $A_1$ and $A_2$ channels are written as

$$R(A_1) = \begin{pmatrix} r_1 & 0 & 0 \\ 0 & r_2 & 0 \\ 0 & 0 & r_3 \end{pmatrix} = \begin{pmatrix} |r_1|e^{i\phi_1} & 0 & 0 \\ 0 & |r_2|e^{i\phi_2} & 0 \\ 0 & 0 & |r_3|e^{i\phi_3} \end{pmatrix}, \tag{4}$$

and

$$R(A_2) = \begin{pmatrix} 0 & 0 & 0 \\ 0 & 0 & r_4 \\ 0 & r_4 & 0 \end{pmatrix} = \begin{pmatrix} 0 & 0 & 0 \\ 0 & 0 & |r_4|e^{i\phi_4} \\ 0 & |r_4|e^{i\phi_4} & 0 \end{pmatrix}. \tag{5}$$

Using the incoming light polarization $\varepsilon_{in} = (0, \cos\theta, \sin\theta)$ and the scattered light polarization $\varepsilon_{out} = (0, \cos\theta, \sin\theta)$, we can calculate the angle variation of a scattering intensity via the relation $I \propto |e_{in} \cdot R \cdot e_{out}|^2$.

The resulting angle-dependent Raman intensities for the $A_1$ and $A_2$ modes in $(b\theta)$ and $(\theta\theta)$ polarizations are

$$I_{A_1}^{(b\theta)} \propto (|r_2|\cos\theta)^2, \tag{6}$$

$$I_{A_2}^{(b\theta)} \propto (|r_4|\sin\theta)^2, \tag{7}$$

$$I_{A_1}^{(\theta\theta)} = \left(|r_2|\cos^2\theta\cos^2\phi_{32} + |r_3|\sin^2\theta\right)^2 + |r_2|^2\cos^4\theta\sin^2\phi_{32}, \quad (8)$$

$$I_{A_2}^{(\theta\theta)} = (r_4\sin2\theta)^2. \quad (9)$$

Here, $\phi_{32} = \phi_3 - \phi_2$ is the phase difference between the Raman tensor elements $r_3$ and $r_2$.

In the $(\theta\theta)$ channel, the angle-dependent $I_{(\theta\theta)}$ of the total magnetic continuum is well described with the parameters $|r_3/r_2| = 0.36$ and $|r_4/r_2| = 0.45$ at $T = 4.3$ K. We find that the $T = 300$ K angular data are fitted with the parameters $|r_3/r_2| = 0.71$ and $|r_4/r_2| = 0.27$. At high temperatures, the increased ratio of $|r_3/r_2|$ against $|r_4/r_2|$ suggests that the $A_1$ symmetry becomes dominant over the $A_2$ symmetry. In a similar manner, $I^{(b\theta)}$ is described with the coefficient ratio of $|r_4/r_2| = 0.59$ at $T = 4.3$ K and 0.37 at $T = 300$ K.

As the magnetic continuum consists of two different excitations, we single out the rotational symmetry of spinons and triplons. In both $(b\theta)$ and $(\theta\theta)$ polarizations, the spinon excitations possess the $A_1$ symmetry and are described by Eq. (6) with the parameters $|r_2| = 1$ and $|r_4| = 0$ and Eq. (8) with $|r_3/r_2| = 0.28$ and $|r_4/r_2| = 0$, respectively. In contrast, the triplon excitations contain the $A_1 + A_2$ symmetry. The angle variation of their intensity is described with the parameters $|r_2| = 0.91$ and $|r_4| = 1$ in the $(b\theta)$ channel and $|r_3/r_2| = 0.32$ and $|r_4/r_2| = 0.98$ in the $(\theta\theta)$ channel.

## Data availability
All relevant data supporting the findings of this study are provided within the paper and its Supplementary Information files. All raw data generated during the current study are available from the corresponding authors on reasonable request.

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

## Acknowledgements
The work at CAU and SKKU was supported by the National Research Foundation (NRF) of Korea (Grant no. 2020R1A2C3012367, 2020R1A5A1016518, and 2019R1A2C1085907).

## Author contributions
The experimental project was conceived by K.-Y.C. together with S.L. S.L. and K.-Y.C. synthesized single crystals and performed magnetic susceptibility measurements. Y.C., J.H.L., S.R.L., and M.-J.S. performed the Raman scattering experiments. Y.C., J.H.L., K.-Y.C., and M.-J.S. analyzed the Raman data. Data analysis and figure preparation were performed by Y.C. and S.L. The manuscript was written through the contributions of all authors.

## Competing interests
The authors declare no competing interests.
