## [Peer Review File · Nature Communications]

Reviewers' Comments:

Reviewer #1:

Remarks to the Author:

The updated manuscript (which was transferred from NatPhys) addresses all comments raised by 3 referees in the previous round of the review in a substantial and informative way. Their experimental findings are quite interesting. The identification of triplon modes from transverse to spin chains polarization channels represents a new step in studies of the dynamical response of low-dimensional frustrated materials. The fact that purported triplon modes move to lower frequency while simultaneously becoming sharper at a temperature below the inter-chain scale J' is quite consistent with the proposed triplon scenario. I think this paper will attract the attention of the research community and recommend its publication in Nature Communications.

Reviewer #2:

Remarks to the Author:

The authors have addressed the issues I raised regarding the ways to interpret experimental data. Although, as the authors point out, many issues are still unclear due to the limited theoretical understanding of what to expect in this system, I believe that the manuscript certainly contains data that is important to the field. The presentation of the paper has also been improved from the previous version. I recommend publication in Nat. Comm., but want to request the author to clarify one more point, that is, the issue with triplon binding below $T = J'$ raised by Reviewer 3. The authors have revised Fig. 4 and the text to point out peak softening and sharpening below $T = J'$. Although I can identify the anomaly and understand in general that bound object would appear sharper when temperatures goes below the binding energy scale, I don't understand why peak energy softening is consistent with this scenario. The authors might want to clarify it if they have an interpretation.

Reviewer #3:

Remarks to the Author:

The manuscript initially was submitted to Nature Physics. I agree with the Referee 1 from the previous review cycle, that although the findings are probably valuable adding onto current understanding of the field, it is still difficult to capture the major novelty of the study (in particular, if to assume that spinons and triplons have been already observed in this materials previously by another group).

The authors claimed that the strong thermal evolution of spinon scattering is comparable with the bosonic spinon scenario. On the other hand, I do not feel that the authors have clear understanding of this observation, raising the question if it is a natural consequence of the lowering spatial symmetry in a TTL state or not.

In addition, I am not convinced by the Author's answer on my previous question regarding the origin of the gap $\Delta_1 \sim 14$ K. Authors claimed that this gap corresponds to the triplon gap. Is that so? As follows from the Nawa et al. (Ref. 29) as well as from the presented data (Fig. 1e), the triplon gap is located at about 45 cm^{-1} , that corresponds to 65 K, but not 14 K.

Based on that I cannot recommend the manuscript for publication in Nature Communications.

Reviewer #1 (Remarks to the Author):

The updated manuscript (which was transferred from NatPhys) addresses all comments raised by 3 referees in the previous round of the review in a substantial and informative way. Their experimental findings are quite interesting. The identification of triplon modes from transverse to spin chains polarization channels represents a new step in studies of the dynamical response of low-dimensional frustrated materials. The fact that purported triplon modes move to lower frequency while simultaneously becoming sharper at a temperature below the inter-chain scale J' is quite consistent with the proposed triplon scenario. I think this paper will attract the attention of the research community and recommend its publication in Nature Communications.

ambiguous.

Authors) We appreciate the Reviewer for his/her positive evaluation of the significance and novelty of our work and for recommending the publication of our work in Nature Communications.

Reviewer #2 (Remarks to the Author):

The authors have addressed the issues I raised regarding the ways to interpret experimental data. Although, as the authors point out, many issues are still unclear due to the limited theoretical understanding of what to expect in this system, I believe that the manuscript certainly contains data that is important to the field. The presentation of the paper has also been improved from the previous version. I recommend publication in Nat. Comm.,

Authors) We thank the Reviewer for his/her positive evaluation of our revised manuscript and for recognizing the significance of our work. We have further improved the presentation style by reflecting the Reviewer's concern.

but want to request the author to clarify one more point, that is, the issue with triplon binding below $T = J'$ raised by Reviewer 3. The authors have revised Fig. 4 and the text to point out peak softening and sharpening below $T = J'$. Although I can identify the anomaly and understand in general that bound object would appear sharper when temperatures goes below the binding energy scale, I don't understand why peak energy softening is consistent with this scenario. The authors might want to clarify it if they have an interpretation.

Authors) Indeed, we omitted to discuss the peak softening of the triplon bound mode. The peak energy of a triplon bound mode (E_{TB}) is given by the difference between twice the energy (E_T) of a bare triplon and the triplon binding energy (E_{BE}): $E_{TB}=2xE_T - E_{BE}$. With lowering temperature, the binding energy increases against thermal energy, thereby reducing E_{TB} . This kind of argument is newly inserted to enhance the readability of our manuscript.

Reviewer #3 (Remarks to the Author):

The manuscript initially was submitted to Nature Physics. I agree with the Referee 1 from the previous review cycle, that although the findings are probably valuable adding onto current understanding of the field, it is still difficult to capture the major novelty of the study (in particular, if to assume that spinons and triplons have been already observed in this materials previously by another group).

The authors claimed that the strong thermal evolution of spinon scattering is comparable with the bosonic spinon scenario. On the other hand, I do not feel that the authors have clear understanding of this observation, raising the question if it is a natural consequence of the lowering spatial symmetry in a TTL state or not.

Authors) We thank the Reviewer for his/her additional comments, which enabled us to improve further the presentation style of our results. We agree with the Reviewer's opinion that the observation of spinons and triplons in ATLS is just a confirmation of the earlier neutron result from a singlet sector. However, it is regrettable for the Reviewer not to acknowledge that our present study conveys new information about dynamics, symmetry, and statistics beyond the simple identification of two kinds of quasiparticles predicted in ATLS. It turns out that the dynamical behaviors of spinons and triplons are much rich in the singlet sector and cannot be described simply in terms of the spinon and triplon dispersion determined by an inelastic neutron study. More singularly, the quasilinear softening of spinon excitations, compatible with the bosonic spinon scenario, forms a key finding that is unprecedented in the thermal evolution of spinons. Within the scope of the present work, we can speculate its underlying mechanism, yet we believe that these unparalleled features will shine new light on quasiparticle's behavior in dimensional reduction and will motivate future investigations, deserving the publication in Nature Communications.

In addition, I am not convinced by the Author's answer on my previous question regarding the origin of the gap $\Delta_1 \sim 14$ K. Authors claimed that this gap corresponds to the triplon gap. Is that so? As follows from the Nawa et al. (Ref. 29) as well as from the presented data (Fig. 1e), the triplon gap is located at about 45 cm^{-1} , that corresponds to 65 K, but not 14 K.

Authors) We thank the Reviewer for spotting this out. As the Reviewer rightly pointed out, in the case of a well-localized triplon excitation, the triplon gap extracted from the activation behavior should be identical to the peak energy of the triplon excitations. Thus, the discrepancy between them means that the triplon excitations are dressed by many-body effects and spinon excitations in a Raman scattering process. Indeed, we observed the broad triplon modes as shown in Fig. 1e and Fig. 4c. As evidenced by our detailed temperature and angular dependence data, this is not a matter of a wrong assignment. Rather, these features showcase that pronounced triplon dynamics observed in a singlet sector cannot be captured within a triplon dispersion observed in a triplet sector. At any rate, to avoid confusion to the reader, we clearly state that an underestimation of the triplon gap is due to the fact that the triplon is dressed by spinons in a spin-conserving Raman scattering process.

Based on that I cannot recommend the manuscript for publication in Nature Communications.

Reviewers' Comments:

Reviewer #3:

Remarks to the Author:

The manuscript looks much more suitable for publication. I would recommend the manuscript for publication now.

Reviewer #3 (Remarks to the Author):

The manuscript looks much more suitable for publication. I would recommend the manuscript for publication now.

Authors) We appreciate the Reviewer for his/her positive evaluation of our revised manuscript and for recommending the publication of our work in Nature Communications.